# Activity Interactions of Crude Biopreservatives against Spoilage Yeast Consortia

**Maxwell Mewa-Ngongang** [1,2,3,*] , **Heinrich W. du Plessis** [1] , **Edwin Hlangwani** [1,2] ,
**Seteno K. O. Ntwampe** [2,3] , **Boredi S. Chidi** [1,2] , **Ucrecia F. Hutchinson** [1,2,3] **and Neil P. Jolly** [1]

1   Post-Harvest and Agro-Processing Technologies, ARC Infruitec-Nietvoorbij (The Fruit, Vine and Wine Institute of the Agricultural Research Council), Private Bag X5026, Stellenbosch 7599, South Africa
2   Bioresource Engineering Research Group (*BioERG*), Department of Biotechnology, Cape Peninsula University of Technology, P.O. Box 652, Cape Town 8000, South Africa
3   Department of Chemical Engineering, Cape Peninsula University of Technology, P.O. Box 652, Cape Town 8000, South Africa
*   Correspondence: mewamaxwell@gmail.com; Tel.: +27-021-809-3442

**Abstract:** It is common to find different spoilage organisms occurring in the same food item, which usually requires food producers to utilize a mixture of synthetic preservatives to control spoilage. This study evaluated the interaction between mixtures of crude biopreservatives against consortia of common spoilage yeasts occurring in beverages. Crude biopreservatives produced from separate yeasts were formulated in different growth inhibition combinations (GICs), i.e., GIC1 (*Candida pyralidae* Y1117 and *Pichia kluyveri* Y1125), GIC 2 (*C. pyralidae* Y1117 and *P. kluyveri* Y1164), GIC3 (*P. kluyveri* Y1125 and *P. kluyveri* Y1164), and GIC4 (*C. pyralidae, P. kluyveri* Y1125 and *P. kluyveri* Y1164). The spoilage yeast consortia combinations, i.e., SC1 (*Dekkera. anomala* and *D. bruxellensis*), SC2 (*D. anomala* and *Zygosaccharomyces bailii*), SC3 (*D. bruxellensis* and *Z. bailii*), and SC4 (*D. anomala*, *D. bruxellensis* and *Z. bailii*), were also prepared. The highest growth inhibition activities of the crude biopreservatives were observed at a pH of 3.0 and 2.0 for *C. pyralidae* and *P. kluyveri* strains, respectively, while reduced activity was observed at a pH of 4.0 and 5.0. The growth inhibition proficiency depended on the spoilage yeast or the consortia of spoilage yeasts. Biocontrol agents from an individual yeast or mixtures can be used to prevent food and beverage spoilage.

**Keywords:** crude biopreservatives; grape pomace; microbial consortia; beverage spoilage yeasts; growth inhibition activity

## 1. Introduction

Protection of food and beverages against microbiological spoilage is essential for maintaining an adequate food supply to growing world populations. Yeast species such as *Dekkera bruxellensis*, *Dekkera anomala*, *Zygosaccharomyces bailii*, *Zygosaccharomyces fermentati*, *Zygosaccharomyces florentinus*, *Zygosaccharomyces microellipsoides*, *Zygosaccharomyces rouxii*, *Hanseniaspora uvarum*, *Candida guilliermondii*, and even *Saccharomyces cerevisiae* have been reported to cause spoilage in alcoholic and non-alcoholic beverages [1–3]. To curb microbial spoilage, synthetic chemicals and/or physical methods can be used [3–7]. Physical methods include pasteurization, cold processing, filtration, the control of water content, ultrasound processing, and irradiation. However, some of these physical methods can have a negative effect on the quality of food items, such as fruits, vegetables, and beverages [3,5–7]. Furthermore, thermophiles, spores, psychrophiles, and xerophiles can sometimes survive these methods [8,9].

Chemical preservatives such as $SO_2$, dimethyl dicarbonate, benzoate, benzoic, lactic, sorbic and acetic acid, triazoles, hydroanilide fenhexamid, dicarboximides, and succinate dehydrogenase sorbate are commonly used to extend the shelf life and to improve the safety of food. However, in some cases, more than one chemical preservative is used to prevent spoilage from a consortium of spoilage microorganisms. This can lead to the presence of undesirably high levels of chemical preservatives in food. Increasingly, consumer fears about potential preservative toxicity and antimicrobial-resistant pathogens in food have led to strict regulations for the use of preservatives on fruit and in fruit-derived beverages. Some regulations also advocate the use of safer alternative preservatives [10–12].

The desire for a safe alternative to synthetic chemicals has led to the investigation of benign microorganisms, specifically yeasts and bacteria, with growth inhibition properties against spoilage microorganisms [1,2,13,14] From bioprospecting studies, some microorganism-derived biocontrol agents, such as killer toxins, have been identified [1,2,15]. Volatile organic compounds (VOCs) such as ethyl acetate, phenylethyl alcohol, 1,3,5,7-cyclooctatetraene, isobutyl acetate, 2-phenyl ethyl acetate, ethyl propionate, isoamyl acetate, and isoamyl alcohol have also been linked to growth inhibition properties [2,15–19].

In previous studies [18,20,21], yeast-derived crude biopreservatives from *C. pyralidae* Y1117, *P. kluyveri* Y1125, and *P. kluyveri* Y1164 were individually tested against selected spoilage fungi and yeasts. However, the growth inhibition efficiency of combining the different crude biopreservatives has not been investigated. The aim of this study was therefore to investigate the growth inhibition activity of individual and mixtures of crude biopreservatives produced by *C. pyralidae* Y1117, *P. kluyveri* Y1125, and *P. kluyveri* Y1164 against individual and consortia of spoilage yeasts.

## 2. Materials and Methods

### 2.1. Yeasts Selection, Growth Inhibition Activity (GIA), and Screening Assays (Plate Assays)

Three strains of yeasts, *Candida pyralidae* Y1117, *Pichia kluyveri* Y1125, and *Pichia kluyveri* Y1164, were obtained from the culture collection of the Agricultural Research Council (ARC Infruitec-Nietvoorbij, Stellenbosch, South Africa). *Candida pyralidae* Y1117 was previously isolated from grape must and *Pichia kluyveri* Y1125 and *Pichia kluyveri* Y1164 were isolated from "Marula" (*Scelerocarya birrea*) fruit juice. These yeasts were screened for growth inhibition activity against the beverage spoilage yeasts listed in Table 1. *Dekkera anomala, Dekkera bruxellensis*, and *Zygosaccharomyces bailii* were used as spoilage yeasts and seeded into grape pomace extract agar (GEA) at the desired concentration, depending on the assay being carried out, as described by Mewa-Ngongang et al. [21]. All plate assays were performed in triplicate using 90 mm petri dishes.

A cross-screening procedure was performed whereby the selected growth inhibiting yeasts were also screened against each other. The grape pomace extract was also tested for growth inhibition activity against the selected spoilage yeasts (*Dekkera anomala, Dekkera bruxellensis*, and *Zygosaccharomyces bailii*. Growth inhibition activity and quantification was carried out as described by Mewa-Ngongang et al. [18,20]. Growth inhibition activity was quantified using the concept of the volumetric zone of inhibition (VZI), with the units being reported as a liter contaminated solidified media per mL crude biopreservatives used (L. CSM/mL BCU), as described by Mewa-Ngongang et al. [20]. This approach was used to quantify what volume of crude biopreservatives would be needed to inhibit the growth of spoilage organisms at a specific concentration in a liter of contaminated solidified media or beverage.

**Table 1.** Screening for growth inhibition activity, including the cross screening of biocontrol yeasts against each other on grape pomace extract. All yeasts used in the study.

| | | Biological control yeasts | | | |
|---|---|---|---|---|---|
| | | *Candida pyralidae* Y1117 | *Pichia kluyveri* Y1125 | *P. kluyveri* Y1164 | Grape Pomace extract |
| Spoilage yeasts | *Dekkera anomala* C96V37 | - | + | + | - |
| | *D. anomala* C96V38 | + | + | + | - |
| | *D. bruxellensis* C96V33 | + | + | + | - |
| | *D. bruxellensis* C96V30 | - | + | + | - |
| | *Brettanomyces lambicus* Y0106 | - | - | - | - |
| | *B. lambicus* Y0175 | + | + | - | - |
| | *B. lambicus* Y0191 | + | + | - | - |
| | *B. lambicus* Y0167 | + | + | - | - |
| | *C. magnoliae* Y1061 | - | - | - | - |
| | *C. guilliermondii* Y0848 | + | - | - | - |
| | *Zygosaccharomyces bailii* Y0070 | + | - | - | - |
| | *Z. bailii* Y0891 | + | - | - | - |
| | *Z. bailii* Y0186 | + | - | - | - |
| | *Z. rouxii* Y0115 | - | - | - | - |
| | *Z. rouxii* Y0111 | - | - | - | - |
| | *Z. microellipsoides* Y0159 | - | - | - | - |
| | *Z. cidri* Y0169 | - | + | + | - |
| | *Z. florentinus* Y0277 | - | - | - | - |
| | *Z. fermentati* Y0182 | - | - | - | - |
| | *Z. bisporus* Y0288 | - | - | - | - |
| | *Z. bisporus* Y0113 | - | - | - | - |
| Biological control yeasts | *C. pyralidae* Y1117 | - | - | - | - |
| | *P. kluyveri* Y1125 | - | - | - | - |
| | *P. kluyveri* Y1164 | - | - | - | - |

## 2.2. Media, Inoculum Preparations, and Crude Biopreservative Production

Grape pomace juice extract was used as the primary fermentation medium in this study. The grape pomace from which the extract was obtained originated from the research cellar of the Agricultural Research Council (ARC Infruitec-Nietvoorbij, Stellenbosch, South Africa). The extraction method used was similar to the normal white grape juice extraction procedure, but with an increased extraction pressure (1 to 2 bar) to allow the recovery of the remaining juice (extract) from the pomace. The juice from the grape pomace was composed of 210 g.L$^{-1}$ total sugar, 185.12 mg L$^{-1}$ yeast assimilable nitrogen (YAN), and 34.13 mg L$^{-1}$ ammonium-nitrogen. The extracted residual juice from Chenin Blanc grape pomace was diluted to different sugar concentrations (150, 100, 75, and 50 g L$^{-1}$ total sugar, equal ratio of glucose and fructose), and adjusted to pH 5.0 using 0.1 M NaOH, subsequent to autoclaving for 20 min at 120 °C, which resulted in grape pomace extract broth (GEB). The availability of yeast assimilable nitrogen (YAN) to support yeast growth was also quantified using an enzyme robot (Arena 20XT; Thermo Fisher Scientific, Vantaa, Finland). The autoclaved GEB was used for yeast growth and crude biopreservative production. Concurrently, the extracted juice from the pomace was supplemented with 2% agar bacteriological and autoclaved, which was then poured on to petri dishes. Hereafter, this medium will be referred to as grape pomace extract agar (GEA). Throughout the study, GEB with a suitable sugar concentration, which best supported microbial growth, was used to grow the inoculum of both the biopreservative-producing yeasts and the beverage spoilage yeasts. For the biopreservative-producing yeasts, the incubation temperature for the inoculum was 25 °C for 24 h in a shaking incubator (150 rmin$^{-1}$). Similar conditions to those described by Mewa-Ngongang et al. [2,20], using GEB as fermentation medium, were used for the production of the crude biopreservatives, whereby samples were withdrawn after 24 h of fermentation and tested for growth inhibition activity.

## 2.3. Preparation of Crude Growth Inhibition Mixtures and Spoilage Yeast Consortia

### 2.3.1. Mixtures of Crude Biopreservatives

*Candida pyralidae* Y1117, *Pichia kluyveri* Y1125, and *Pichia kluyveri* Y1164 were grown separately in GEB (75 g L$^{-1}$) for 24 h at 25 °C in a shaking incubator (LM-53OR, RKC® Instrument INC, Ohta-ku

Tokyo, Japan) at 150 r/min. The broths, after fermentation, were centrifuged for 5 min at 5000 r min$^{-1}$. The resulting supernatant from each of the yeast cultures was aliquoted into Eppendorf tubes (2 mL). The mixture of the crude biopreservatives was prepared as follows: a volume (50 μL) from each of the three cell free supernatants was mixed in a separate Eppendorf tube (2 mL) and stored at 4 °C for further use. The different growth inhibitor combinations (GICs) from the cell free supernatants were GIC1 (*C. pyralidae* Y1117 and *P. kluyveri* Y1125), GIC2 (*C. pyralidae* Y1117 and *P. kluyveri* Y1164), GIC3 (*P. kluyveri* Y1125 and *Pichia kluyveri* Y1164), and GIC4 (*C. pyralidae* Y1117, *P. kluyveri* Y1125, and *P. kluyveri* Y1164).

### 2.3.2. Spoilage Yeast Consortia Preparation

Three beverage spoilage yeast strains, *Dekkera anomala*, *Dekkera bruxellensis*, and *Zygosaccharomyces bailii*, were grown in GEB without agitation for 48 h at 25 °C prior to the growth inhibition assays. From the 48-h-old spoilage yeast cultures, a volume (2 mL) of the broth was centrifuged for 5 min at 5000 r min$^{-1}$, after which the supernatant was discarded, with the resulting pellet from each of the spoilage yeast cultures being washed with sterile distilled water prior to storage for the assays. The mixtures of spoilage yeasts were prepared in different combinations, i.e., SC1 (*D. anomala* and *D. bruxellensis*), SC 2 (*D. anomala* and *Z. bailii*), SC3 (*D. bruxellensis* and *Z. bailii*), and SC4 (*D. anomala*, *D. bruxellensis*, and *Z. bailii*). For each consortium, the individual spoilage organism was at a final concentration of $10^3$ cells/mL. The prepared mixtures were then used to seed the GEA plated for growth inhibition assays.

### 2.4. Effect of Proteolytic Enzymes on the Denaturation of Crude Biopreservatives

To evaluate the nature of the crude biopreservatives, it was important to subject the crude samples to protease treatments in order to determine whether the crude biopreservatives responsible for growth inhibition activity were of a protein nature. The assay used was adapted from Mehlomakulu et al. [2]. The protease enzymes used were Proteinase K, pepsin, and proteases from *Aspergillus saitoi* and *Rhizopus* spp. (Sigma-Aldrich, Darmstadt, Germany). Thereafter, treated crude samples were tested for growth inhibition activity, using the method described by Mew-Ngongang et al. [20], with the positive control being used as the crude sample that was not subjected to any of the treatments. Furthermore, the negative control consisted of just the protease without the crude biopreservative, with all samples being prepared in three replicates.

### 2.5. Effect of pH and Temperature on Activity and Stability of Crude Biopreservatives

The temperature activity study was performed by spotting three replicates, with 20 μL of the crude sample in a 5 mm diameter well created on GEA seeded with *D. anomala* as a spoilage organism. The plates were incubated at 5, 15, 20, 25, 30, and 40 °C, respectively. Additionally, the pH activity was also determined by spotting 20 μL on GEA adjusted to a pH of 2.0, 3.0, 4.0, and 5.0 respectively, which is the pH range of many foods and beverages. These GEA plates were then incubated at 25 °C until the volumetric zone of inhibition (VZI) was observed. Thereafter, the growth inhibition activity was measured. A stability test was also performed after confirming the temperature and pH optima. The stability test was carried out by storing the crude biopreservative mixtures at different temperatures (−10, 5, 15, 20, 25, 30, and 40 °C) for 16 weeks. A volume of 20 μL of the respective stored samples was spotted onto three replicate plates prepared at the pH optima, subsequent to incubation at the temperature optima until the zone of inhibition was observed.

### 2.6. Growth Inhibition Study of Crude Biopreservative Mixtures against Spoilage Yeast Consortia

### 2.6.1. Effect of Cell Free Supernatants on Growth Inhibition Activity of Single Spoilage Organism

The growth inhibition efficiency of *C. pyralidae* Y1117, *P. kluyveri* Y1125, and *P. kluyveri* Y1164 extracellular metabolites was tested by spotting 20 μL of the cell free supernatant in a 5 mm diameter

well on GEA seeded with individual spoilage yeasts at the concentration of $10^6$ cells/m. All treatments had three replicates and the plates were incubated at 25 °C until zones of inhibition were observed around the wells of the GEA plates. The growth inhibition activity was quantified as described by Mewa-Ngongang et al. [20].

2.6.2. Effect of Cell Free Supernatants from Single Yeasts on Growth Inhibition Activity of Spoilage Yeast Consortia

The ability of *C. pyralidae* Y1117, *P. kluyveri* Y1125, and *P. kluyveri* Y1164 extracellular metabolites to inhibit the consortia of spoilage yeasts was assessed by spotting 20 μL of the crude biopreservation samples in the GEA plate wells prepared according to the different spoilage combinations. The spoilage combination plates were also prepared with three replicates. The plates were incubated at 25 °C until the zone of inhibition was obtained. The growth inhibition quantification was also assessed by the method developed and reported by Mewa-Ngongang et al. [20].

2.6.3. Effect of Mixed Crude Biopreservatives on the Growth Inhibition of Single and Consortia of Spoilage Organisms

A mixture of extracellular crude biopreservatives from the three producing yeasts was prepared by mixing an equivalent volume of the crude supernatants. The spoilage yeast consortia were prepared by mixing an equivalent volume of each culture ($10^3$ cells/mL) of the individual spoilage yeasts. From the mixture of the crude supernatants, 20 μL was spotted on different spoilage yeast consortia plates, prepared in three replicates, followed by incubation, until zones of inhibition were observed, i.e., when quantifiable.

## 3. Results and Discussion

### 3.1. Growth Inhibition Activity Screening and Crude Pounds Production

The growth inhibition activity of *C. pyralidae* Y1117, *P. kluyveri* Y1125, and *P. kluyveri* Y1164 was screened against the yeasts listed in Table 1, which included *D. anomala* (2), *D. bruxellensis* (2), *Brettanomyces lambicus* (4), *C. magnoliae*, *C. guilliermondii*, *Z. baillii* (3), *Z. bisporus* (2), *Z. cidri*, *Z. fermentati*, *Z. florentinus*, *Z. microellipsoides*, and *Z. rouxii* (2). A zone of inhibition around the sensitive yeast colony indicates growth inhibition as depicted in Figure 1. In this study, in order to limit uncertainties in the origin of the growth inhibition activities, biopreservative-producing yeasts were screened against the spoilage yeasts and were cross-screened among themselves to see whether they displayed inhibition activity against each other (Table 1). The results show that the biopreservative-producing yeasts exhibited no growth inhibition activity against each other. The GP extracts did not have any growth inhibition effect on any of the yeasts studied.

Thereafter, grape pomace juice extract broth was used as the fermentation medium for the production of crude biopreservatives. This was confirmed by the zone of inhibition observed on the plates when 20 μL of the crude samples from each yeast culture was spotted in a 5 mm well on the GEA plates against spoilage organisms. The VZI was not quantified in this section because only visual inspections were required for further experiments.

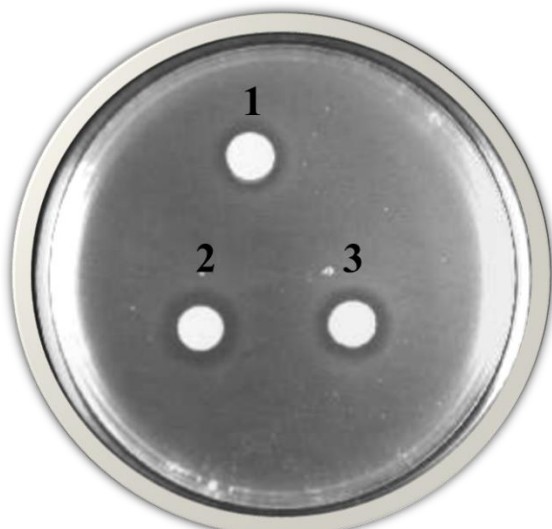

**Figure 1.** Growth inhibition activity of *Candida pyralidae* Y1117 (1), *Pichia kluyveri* Y1125 (2), and *Pichia kluyveri* Y1164 (3) against *Dekkera anomala* on grape pomace extract agar. One plate out of all the different treatments was selected for a visual representation of growth inhibition activity.

### 3.2. Effect of Proteolytic Enzymes on the Denaturation of the Crude Biopreservatives

The denaturation ability of crude biopreservatives was investigated using a proteolytic enzymes treatment approach. Minimal denaturation of the crude biopreservatives was observed (data not shown). This suggests that the crude biopreservatives responsible for growth inhibition activity are not proteinaceous compounds produced during fermentation. A few *Candida* species, including *Candida pyralidae*, have been found to secrete non-proteinaceous type killer toxins that have growth inhibition activity [2]. This study therefore suggests that these yeasts could offer different types of metabolites for growth inhibition, against a variety of spoilage organisms, than those of a proteomic nature, which is in agreement with other reports [15,22]. We previously reported [18] that the growth inhibition activity of the crude biopreservatives produced by *C. pyralidae* Y1117, *P. kluyveri* Y1125, and *Pichia kluyveri* Y1164 was possibly due to volatile organic compounds, i.e., isoamyl acetate, isoamyl alcohol, 2-phenyl ethylacetate, and 2-phenyl ethanol. However, the focus of this study was on investigating the growth inhibition activity of the crude biopreservatives, which can potentially be used as pre- and post-harvest biocontrol agents.

### 3.3. Effect of pH and Temperature on Activity and Stability of Crude Biopreservatives

The effect of pH and temperature on activity was tested. Observations of *C. pyralidae* indicate that the best pH to retain the highest growth inhibition activity was pH 3.0; however, for the two *P. kluyveri* strains, the suitable pH level was 2.0, while a pH of 4.0 and 5.0 showed extremely reduced growth inhibition activity for these yeasts (Figure 2). Similar results were observed against all three spoilage yeasts. Comparing these results to some findings reported in the literature, these pH values are within the range previously reported for other yeasts [2,15,22]. The stability of crude biopreservatives at the pH optima observed was investigated further.

For the stability test of the crude samples at different pH levels, it was noted that *P. kluyveri* samples were stable at a pH of 2.0 for 16 weeks and *C. pyralidae* samples was stable at a pH of 3.0 for the same period. Furthermore, temperature variation resulted in stability assessments for all the crude samples, which were found to be stable between - 10, 5, and 10 °C, respectively, where the activity was retained for more than 16 weeks. The pH and temperature assessments with regard to retaining growth inhibition activity of the crude biopreservatives were performed under different pH and temperature conditions that are usually used for consumer-packaged goods. The optimal activity and stability of the crude biopreservatives were obtained at a lower pH (2 and 3), which falls within

the pH ranges of most beverages. These findings further expand and elucidate the in-situ application of crude biopreservatives in other types of food and beverage items with the same pH and storage temperature conditions.

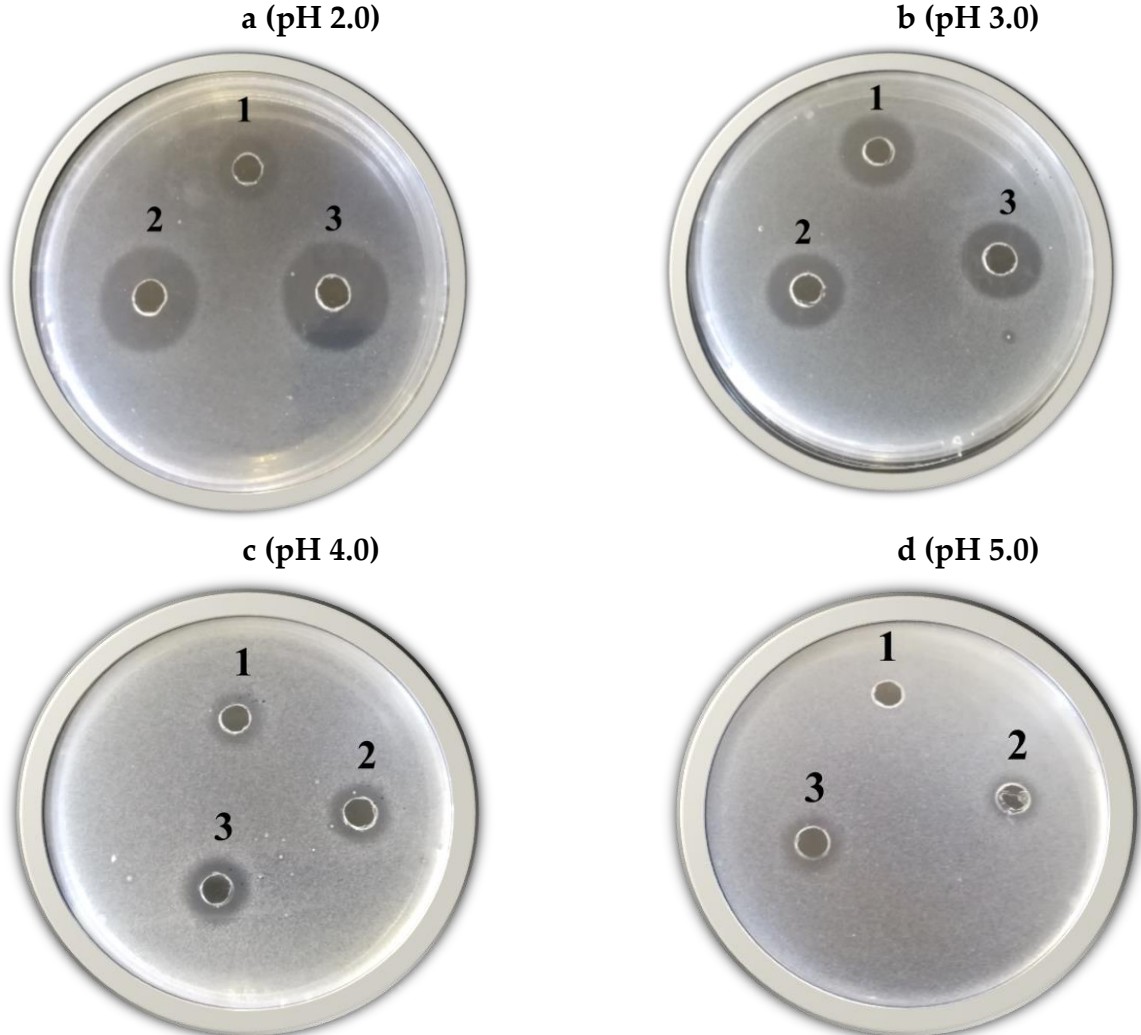

**Figure 2.** Growth inhibition activity of the crude biopreservatives from *Candida pyralidae* Y1117 (1), *Pichia kluyveri* Y1125 (2), and *Pichia kluyveri* Y1164 (3) against *Dekkera anomala* on grape pomace extract agar at different pH levels: (**a**) pH 2.0, (**b**) pH 3.0, (**c**) pH 4.0, and (**d**) pH 5.0.

### 3.4. Growth Inhibition Interactions

3.4.1. Effect of Single Isolates and Their Cell Free Supernatants on Growth Inhibition Activity of Single Spoilage

The quantification results of the effect of single crude versus single spoilage showed that *C. pyralidae* did not inhibit the growth of *D. anomala*, but inhibited the growth of *D. bruxellensis* and *Z. bailii* (Figure 3a). Meanwhile, the *P. kluyveri* strains inhibited the growth of *D. anomala* and *D. bruxellensis* with minimal growth inhibition activity against *Z. bailii*. Growth inhibition activity (visually indicated VZI) of the three biopreservation yeasts against *D. bruxellensis* is shown in Figure 3b.

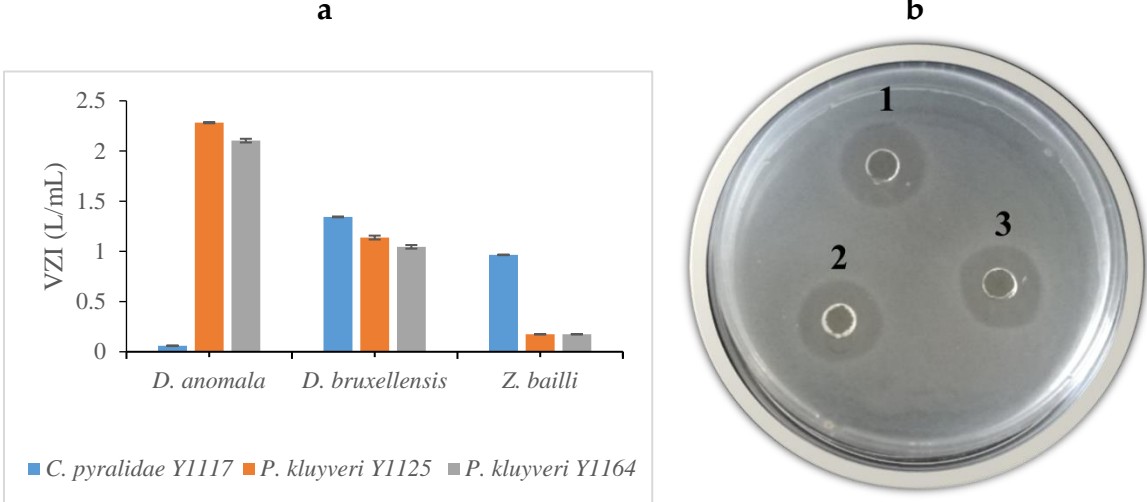

**Figure 3.** (**a**) Values of the volumetric zone of inhibition (VZI) obtained from the single isolates and their cell free supernatants against spoilage yeasts (*Dekkera anomala*, *D. bruxellensis*, and *Zygosaccharomyces bailii*) (**b**) An example of a plate showing the VZI obtained for *Candida pyralidae* Y1117 (1), *Pichia kluyveri* Y1125 (2), and *Pichia kluyveri* Y1164 (3) against *Dekkera bruxellensis.* The values presented are the means of three replicates with the standard deviation ranging between 0.004 and 0.09.

The results obtained were in agreement with what usually occurs industrially during food preservation—microbial spoilage may still occur even though a specific food environment contains a preservation agent. In this case, the occurrence of spoilage organisms could be attributed to the resistance of spoilage organisms to the synthetic preservation agent used. Furthermore, a specific preservation agent could be used in targeting specific spoilage organisms; however, it may happen that a different spoilage organism contaminates the manufactured product. When this phenomenon occurs, the efficacy of the preservation agent is retained, but the biocontrol efficiency may seem lost due to the incorrect microbial target or microbial resistance to the crude biopreservatives used by the spoilage organism. Based on these reasons, more than one preservation agent can be used in the same food/beverage environment; hence the preference to use a mixture of preservation agents. In combating beverage spoilage suspected to occur due to the presence of *D. anomala*, *D. bruxellensis*, and *Z. bailli*, a combination of crude biopreservatives from *Candida pyralidae* Y1117 and *Pichia kluyveri* Y1125 or *Pichia kluyveri* Y1164 is recommended. Furthermore, the high efficacy observed for *P. kluyveri* against *D. anomala* and *D. bruxellensis* means that either *P. kluyveri* Y1125 or Y 1164 could be used in combating spoilage for either single or mixed spoilage organisms.

### 3.4.2. Growth Inhibition Activity of Crude Biopreservatives from Individual Yeasts against Consortia of Spoilage Yeast

Since there was a broad-spectrum growth inhibition activity from single yeasts, the efficacy of the biopreservation extracts against the consortia of spoilage yeasts was assessed. All crude biopreservation samples showed growth inhibition activity against the spoilage yeast consortia, SC1-SC4, albeit, at different efficacy levels (Figure 4a), proving the hypothesis that crude biopreservatives from a single yeast can act against a consortium of spoilage organisms. However, the highest growth inhibition was observed with *C. pyralidae* and *P. kluyveri* Y1164 against SC1 and SC3 (Figure 4b). Figure 4b shows the clear zones of inhibition obtained. This observation suggested that a beverage that is contaminated with a mixture of *D. anomala* and *D. bruxellensis* including a mixture of *D. bruxellensis* and *Z. bailli*, can be preserved using biopreservative from either *C. pyralidae* or *P. kluyveri* Y1164. Although the biopreservatives from *P. kluyveri* Y1125 can also be used in all spoilage organism combinations, a higher dose might be required compared to the quantity used if *C. pyralidae* or *P. kluyveri* Y1164 are to be used as fermenters of the crude biopreservatives of interest. Overall, it can be mentioned that each of the

biopreservative crude samples was able to inhibit the growth of *D. anomala*, *D. bruxellensis*, and *Z. bailli* simultaneously under the same medium and conditions. Previously, *C. pyralidae* yielded a relatively lower growth inhibition activity (VZI) of 1.05 L mL$^{-1}$ when Yeast Peptone Dextrose (YPD) was used as a fermentation media by Mewa-Ngongang et al. [20]. This study suggests that utilizing a combination of crude biopreservatives from different yeasts could improve the inhibition activity.

**a** **b**

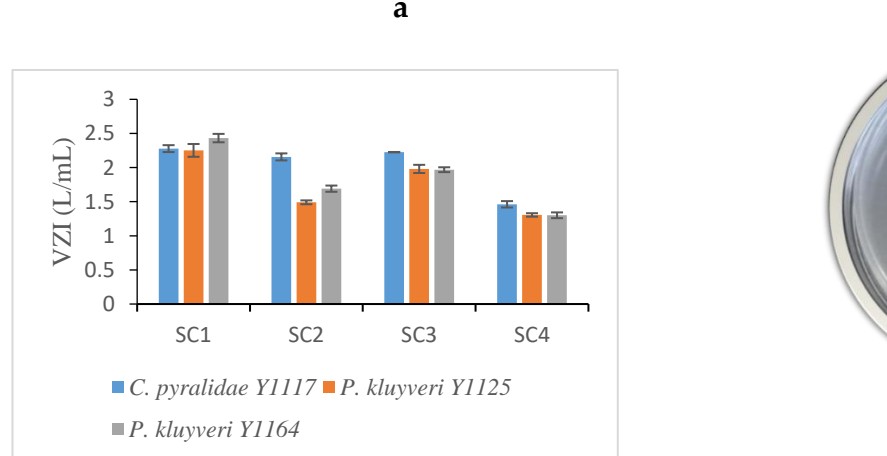 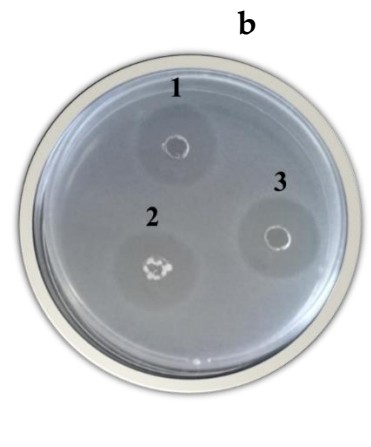

**Figure 4.** (**a**) Growth inhibition activity, expressed as the volumetric zone of inhibition (VZI), of the crude biopreservatives from single yeasts against spoilage yeast consortia (SC) on grape extract agar. (**b**) An example of a plate showing the VZI obtained for *Candida pyralidae* Y1117 (1), *Pichia kluyveri* Y1125 (2), and *Pichia kluyveri* Y1164 (3). SC1 (*D. anomala* and *D. bruxellensis*), SC2 (*D. anomala* and *Z. bailii*), SC3 (*D. bruxellensis* and *Z. bailii*), and SC4 (*D. anomala*, *D. bruxellensis*, and *Z. bailii*). The values are the means of three replicates with the standard deviation ranging between 0.004 and 0.09.

### 3.4.3. Effect of Crude Biopreservative Mixtures on the Growth Inhibition of Single Spoilage Yeasts

A further interest in assessing the efficacy of mixed crude biopreservatives and their ability to inhibit single spoilage yeast was developed. It was observed that the spoilage yeast consortia were a lot more sensitive to the mixed crude biopreservative mixtures. It was found that, in all the growth inhibition combinations (GIC) studied, *D. anomala* was the most sensitive spoilage yeast, symbolized by a bigger VZI of 2.5 L mL$^{-1}$ (Figure 5a), while *D. bruxellensis*, and *Z. bailli* had the least sensitivity, with a VZI of 0.17 L mL$^{-1}$ for GIC3 (Figure 5a,b). A VZI of 2.5 L mL$^{-1}$ suggests that 1 mL of the crude biopreservatives inhibits the growth of *D. anomala* in a 2.5 L of beverage contaminated at a cell concentration of $10^6$ cells mL$^{-1}$. In addition, the least sensitive, i.e., with a VZI of 0.17 L mL$^{-1}$, meant that the same volume (1 mL) of crude biopreservatives can only inhibit the growth of *D. bruxellensis* and *Z. bailli* in a 0.17 L volume of beverage contaminated with $10^6$ cells mL$^{-1}$. The model of the current study therefore showed that crude volumes are dependent on the sensitivity of the spoilage organism towards the biopreservation agents. Although different levels of growth inhibition were observed, it was interesting to note that, if the mixed crude biopreservatives from all yeasts are used, the growth inhibition of spoilage will occur under a minimal biopreservative dosage. When processing food or beverages, the addition of preservatives should always be minimal, while considering an optimum efficacy. This is employed to limit the negative impact of large quantities of the crude biopreservatives in the final product that could affect the taste, texture, and possibly the appearance of the final product. However, based on our observations, a mixture of crude biopreservatives can also be used to target a single spoilage organism.

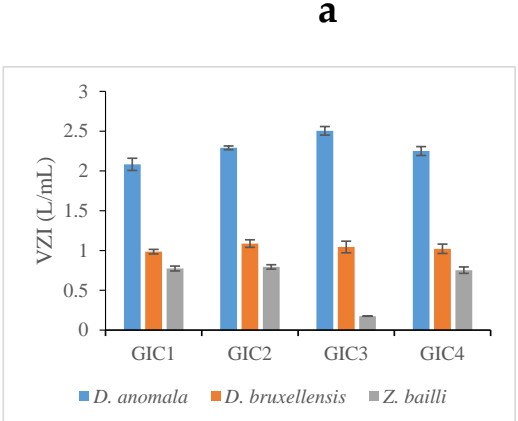
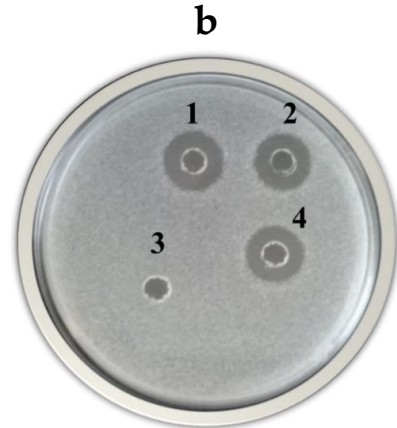

**Figure 5.** (**a**) Values of the volumetric zone of inhibition (VZI) obtained from mixtures of growth inhibition combinations (GICs) against spoilage yeasts (*Dekkera anomala, D. bruxellensis*, and *Zygosaccharomyces bailii*). (b) An example of a plate showing the VZI obtained for GIC1 (*C. pyralidae* Y1117 and *P. kluyveri* Y1125) (1), GIC2 (*C. pyralidae* Y1117 and *P. kluyveri* Y1164) (2), GIC3 (*P. kluyveri* Y1125 and *Pichia kluyveri* Y1164) (3), and GIC4 (*C. pyralidae* Y1117, *P. kluyveri* Y1125, and *P. kluyveri* Y1164) (4) against *Zygosaccharomyces bailli* on grape pomace extract agar. The values presented are means of three replicates with the standard deviation ranging between 0.004 and 0.09.

3.4.4. Effect of Crude Biopreservative Mixtures on the Growth of Spoilage Yeast Consortia

In food spoilage, it is reasonable to hypothesize that different spoilage organisms can be within the same beverage. The efficacy of the crude biopreservative mixture was assessed against consortia of spoilage yeasts. Therefore, it was observed that all growth inhibition combinations (GICs) showed growth inhibition activity at different levels, depending on the spoilage combinations assessed. The highest growth inhibition activity, indicated as the volumetric zone of inhibition (VZI), was observed for GIC2 (2.27 L mL$^{-1}$) and GIC4 (2.21 L mL$^{-1}$) against SC3 (Figure 6a). All GICs showed low growth inhibition activity against SC4, with values of 1.55, 1.41, 1.34, and 1.36 for GIC1, GIC2, GIC3, and GIC4, respectively. This could be due to crude biopreservative interactions in the crude biopreservation mixtures.

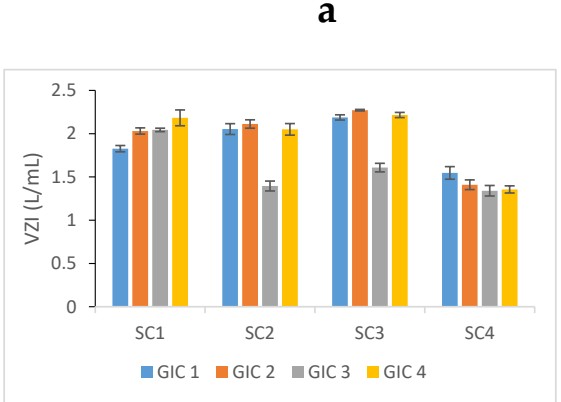
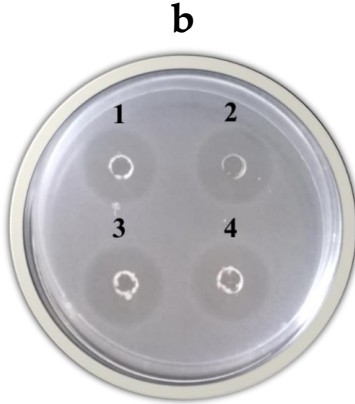

**Figure 6.** (**a**) Values of the volumetric zone of inhibition (VZI) obtained from growth inhibition combinations (GICs), i.e., GIC1, *C. pyralidae* Y1117 and *P. kluyveri* Y1125 (1); GIC2, *C. pyralidae* Y1117 and *P. kluyveri* Y1164 (2); GIC3, *P. kluyveri* Y1125 and *Pichia kluyveri* Y1164 (3); and GIC4, *C. pyralidae* Y1117, *P. kluyveri* Y1125, and *P. kluyveri* Y1164 (4) against spoilage yeast consortia (SC), i.e., SC1 (*D. anomala* and *D. bruxellensis*), SC2 (*D. anomala* and *Z. bailii*), SC3 (*D. bruxellensis* and *Z. bailii*), and SC4 (*D. anomala*, *D. bruxellensis*, and *Z. bailii*). (**b**) An example of a plate showing the VZI obtained from the GICs against SCs. The values presented are the means of three replicates with the standard deviation ranging between 0.004 and 0.09.

In a review by Kumar and Jagadeesh [23], several microbial consortia against phytopathogens were highlighted. The microbial consortia reported thus far against *Botrytis*, *Colletotrichum*, *Rhizoctoniasolani*, and *Pyriculariaoryzae* species were composed of two or more species of yeasts and/or bacteria acting on wounded fruits [23–27].

Comparing the results obtained from this work with those reported on phytopathogens, it could be suggested that microbial consortia cannot only be composed of two strains of microorganisms. Furthermore, this study also suggests that crude biopreservatives from yeasts, without supplementation by chemical preservatives, could be effective against beverage spoilage yeasts, even when they are in a consortium.

## 4. Conclusions and Recommendations

Subsequent to finding the conditions where the crude biopreservatives were stable, the activity of the individual, as well as GICs, was dependent on the composition of the SCs. Our findings showed the potential of GICs from *C. pyralidae* Y1117, *P. kluyveri* Y1125, and *P. kluyveri* Y1164, as an alternative to synthetic chemicals as preservatives in food, including beverages susceptible to contamination by *D. bruxellensis*, *Z. baillii*, and *D. anomala*. Future research should focus on testing the reported GICs against other spoilage microorganisms and toxicological studies should be performed to assess the impact of these GICs on human health, as well as the microorganisms associated with the human gastrointestinal tract.

**Author Contributions:** Conceptualization, M.M.-N. and H.W.d.P.; methodology and software, M.M.-N., S.K.O.N., and B.S.C.; validation, S.K.O.N., H.W.d.P., B.S.C., and N.P.J.; formal analysis, M.M.-N., E.H., and U.F. Hutchinson; investigation, M.M.-N. and U.F.H.; resources, N.P.J., H.W.d.P., and S.K.O.N.; data curation, B.S.C., M.M.-N., and E.H.; writing—original draft preparation, M.M.-N.; visualization, review and editing, S.K.O.N., H.W.d.P., N.P.J., B.S.C., E.H., and U.F.H.; supervision, S.K.O.N., H.W.d.P., and N.P.J.; project administration, N.P.J., H.W.d.P., and S.K.O.N.; funding acquisition, N.P.J., H.W.d.P., and S.K.O.N.

**Funding:** Financial support was provided by the National Research Foundation (NRF) and the Agricultural Research Council (ARC) of South Africa.

**Acknowledgments:** The authors would like to thank the Agricultural Research Council, the National Research Foundation (NRF), and the Bioresource Engineering Research Group (BioERG), Department of Biotechnology, Cape Peninsula University of Technology for funding and infrastructure. The authors thank all students, interns, technicians, and research assistants for their contribution.

**Conflicts of Interest:** The authors hereby declare that they have no conflict of interest.

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
