# Peer review of "Activity Interactions of Crude Biopreservatives against Spoilage Yeast Consortia"

_fermentation, doi:10.3390/fermentation5030053_

Reviewer 1 Report

The article written by Ngongang et al 2019 “Activity interaction of crude biopreservation compounds against beverage spoilage consortia” is significant in today’s world as there is a serious problem of food spoilage. Food spoilage is a major issue for the food industry, leading to food waste, substantial economic losses for manufacturers and consumers, and a negative impact on brand names. Today, food losses are a major concern worldwide especially with an ever-growing world population and the fact that approximately one-third of all food produced for human consumption is either lost or wasted.

In this research, the results showed a variation in growth inhibition proficiency depending on the spoilage organisms or the consortia of spoilage organisms being deactivated. It was then suggested that, a food environment contaminated with a consortium of spoilage organisms can be controlled by employing either the crude biopreservation compounds from individuals yeast or from yeast combinations.

The research addresses how the beverage industries can reduce the contaminations and give the maximum preservation from the contaminating organisms.

Comments

1.      Previous studies related to prevention of beverage contaminations has been missing in the introduction section of the manuscript.

2.      Spoilage yeasts reported in this research is only three which may be more for grapes fermentations but less for other beverage fermentations. I recommend at least five spoilage yeast and few other organism be included in this research for healthy outcome.

3.      What are the different metabolites other than proteomic nature responsible for growth inhibition?

4.      The findings are interesting but is this finding more preferable then the use of chemical preservatives currently used. How about the changes in gut micro flora in human gut after the consumption is an important and serious matter.

Author Response

Response to Reviewer 1 Comments

The article written by Ngongang et al 2019 “Activity interaction of crude biopreservation compounds against beverage spoilage consortia” is significant in today’s world as there is a serious problem of food spoilage. Food spoilage is a major issue for the food industry, leading to food waste, substantial economic losses for manufacturers and consumers, and a negative impact on brand names. Today, food losses are a major concern worldwide especially with an ever-growing world population and the fact that approximately one-third of all food produced for human consumption is either lost or wasted.

In this research, the results showed a variation in growth inhibition proficiency depending on the spoilage organisms or the consortia of spoilage organisms being deactivated. It was then suggested that, a food environment contaminated with a consortium of spoilage organisms can be controlled by employing either the crude biopreservation compounds from individuals yeast or from yeast combinations.

The research addresses how the beverage industries can reduce the contaminations and give the maximum preservation from the contaminating organisms.

Comments

Point 1. Previous studies related to prevention of beverage contaminations has been missing in the introduction section of the manuscript.

Response: Other methods and studies associated with the prevention of beverage contamination has been added to the introduction with the related references. (See line 36-41 under introduction)

Point 2. Spoilage yeasts reported in this research is only three which may be more for grapes fermentations but less for other beverage fermentations. I recommend at least five spoilage yeast and few other organism be included in this research for healthy outcome.

Response: Thank you for the valuable comment. Initially, the biopreservatives producing yeasts were screened against many spoilage organisms. Based on the scope of the proposal submitted and also based on the fact that it was imperative to first assess if those yeasts will be able to produce extracellular compounds with growth inhibition properties, only those three spoilage yeasts were selected. However, the recommendations for future work from this phase will include broadening the scope to include many other spoilage organisms, including bacteria.

Point 3.What are the different metabolites other than proteomic nature responsible for growth inhibition?

Response: There have been many findings regarding the antagonistic characteristics of yeasts, enzymes such as Laminarinases, chitinases and some killer toxins have been found to be produced by yeasts. Given the fact that this work was focused on other antagonistic characteristics other than the aforementioned, we looked at diffusible volatile organic compounds which can act against beverage spoilage organisms. In the context of natural biological control, the antagonistic action mechanisms of some yeasts have been linked to the production of VOCs in the category of alcohols, organic acids and esters to be used in the fruit and beverage industries.

Point 4. The findings are interesting but is this finding more preferable then the use of chemical preservatives currently used. How about the changes in gut micro flora in human gut after the consumption is an important and serious matter.

Response: Thank you for the comment, it is very valuable. This phase of the project formed part of a bigger umbrella project; although the toxicology study was not included in this study, it was recommended as part of future work. (See line under recommendations)

Reviewer 2 Report

Comments:

This manuscript “Activity interaction of crude biopreservation compounds against beverage spoilage consortia.” aimed at the evaluation of the interaction between mixtures of crude biopreservatives against consortia of spoilage organisms occurring in beverages.

The authors achieved this goal by producing the crude biopreservation compounds separately from yeasts and then formulating growth inhibition combinations.

The theme is interesting and very current. The bibliographic review focuses the theme in question.

The authors already have some publications within the theme in question:

1.      Foods. 2019 Feb 2;8(2). pii: E51. doi: 10.3390/foods8020051. Grape Pomace Extracts as Fermentation Medium for the Production of Potential Biopreservation Compounds.

2.      BIOPRESERVATIVES FROM YEASTS WITH ANTIMICROBIAL ACTIVITY AGAINST COMMON FOOD, AGRICULTURAL PRODUCE AND BEVERAGE SPOILAGE ORGANISMS. June 2017, In book: Antimicrobial Research: Novel bioknowledge and educational programs. Editors: A. Méndez-Vilas.

So, they have expertise, and really can develop a good work.

However, concerning this article (Activity interaction of crude biopreservation compounds against beverage spoilage consortia), I have some recommendations that I a sure will improve it.

1.      In the paragraph “The quantification results of the effect of single crude versus single spoilage showed that C. pyralidae did not inhibit the growth of D. anomala, but inhibited that of D. bruxellensis and Z. bailii (Figure 2) “ it should be “The quantification results of the effect of single crude versus single spoilage showed that C. pyralidae did not inhibit the growth of D. anomala, but inhibited that of D. bruxellensis and Z. bailii (Figure 3 a)…The Figure number is not correct!

2.       In the Figure 4 caption it should be mentioned the spoilage combination: SC 1 (D. anomala and D. bruxellensis); SC 2 (D. anomala and Z. bailii), SC 3 (D. bruxellensis and Z. bailii); and SC 4 (D. anomala, D. bruxellensis and Z. bailii), because it would be easier to understand the figure.

3.       The same goes for Figure 5, the different growth inhibitor combinations (GICs) composition should be mentioned - GIC 1 (C. pyralidae Y1117 and P. kluyveri Y1125); GIC 2 (C. pyralidae Y1117 and P. kluyveri Y1164), GIC 3 (P. kluyveri Y1125 and Pichia kluyveri Y1164); GIC 4 (C. pyralidae Y1117, P. kluyveri Y1125 and P. kluyveri Y1164).

4.       I also think that with an ANOVA data treatment (statistical analysis) should be performed for each experiment since, for instances, in Figure 4, different results were obtained. Is clearly visible in the graphic (figure 4 a), however, are those results statistical significant?

So, in my opinion, the article can be improved.

Author Response

Response to Reviewer 2 Comments

Comments:

This manuscript “Activity interaction of crude biopreservation compounds against beverage spoilage consortia.” aimed at the evaluation of the interaction between mixtures of crude biopreservatives against consortia of spoilage organisms occurring in beverages.

The authors achieved this goal by producing the crude biopreservation compounds separately from yeasts and then formulating growth inhibition combinations.

The theme is interesting and very current. The bibliographic review focuses the theme in question.The authors already have some publications within the theme in question:

1. Foods. 2019 Feb 2;8(2). pii: E51. doi: 10.3390/foods8020051. Grape Pomace Extracts as Fermentation Medium for the Production of Potential Biopreservation Compounds.

2.BIOPRESERVATIVES FROM YEASTS WITH ANTIMICROBIAL ACTIVITY AGAINST COMMON FOOD, AGRICULTURAL PRODUCE AND BEVERAGE SPOILAGE ORGANISMS. June 2017, In book: Antimicrobial Research: Novel bioknowledge and educational programs. Editors: A. Méndez-Vilas.

So, they have expertise, and really can develop a good work.

However, concerning this article (Activity interaction of crude biopreservation compounds against beverage spoilage consortia), I have some recommendations that I a sure will improve it.

Point 1. In the paragraph “The quantification results of the effect of single crude versus single spoilage showed that C. pyralidae did not inhibit the growth of D. anomala, but inhibited that of D. bruxellensis and Z. bailii (Figure 2) “ it should be “The quantification results of the effect of single crude versus single spoilage showed that C. pyralidae did not inhibit the growth of D. anomala, but inhibited that of D. bruxellensis and Z. bailii (Figure 3 a)…The Figure number is not correct!

Response: Thank you for the comment, it has now been corrected (see under ‘Growth inhibition interaction’ line 249-252)

Point 2. In the Figure 4 caption it should be mentioned the spoilage combination: SC 1 (D. anomala and D. bruxellensis); SC 2 (D. anomala and Z. bailii), SC 3 (D. bruxellensis and Z. bailii); and SC 4 (D. anomalaD. bruxellensis and Z. bailii), because it would be easier to understand the figure.

Response: It has now been corrected. See figure 4, 5 and 6 legends.

Point 3. The same goes for Figure 5, the different growth inhibitor combinations (GICs) composition should be mentioned - GIC 1 (C. pyralidae Y1117 and P. kluyveri Y1125); GIC 2 (C. pyralidae Y1117 and P. kluyveri Y1164), GIC 3 (P. kluyveri Y1125 and Pichia kluyveri Y1164); GIC 4 (C. pyralidae Y1117, P. kluyveri Y1125 and P. kluyveri Y1164).

Response: all the abbreviation pertaining to all the growth inhibition combinations were written out in full under figure 5 caption.  

Point 4. I also think that with an ANOVA data treatment (statistical analysis) should be performed for each experiment since, for instances, in Figure 4, different results were obtained. Is clearly visible in the graphic (figure 4 a), however, are those results statistical significant?

 Response: Yes the results found are statistically significant. The values presented in each graph are the average of three replicates with their respective standard deviations found to be statistically significant.

So, in my opinion, the article can be improved.

Round  2

Reviewer 1 Report

The authors are successful to get the results that they have presented here. This make much sense to the project and scientific community but still I find important experiments lacking like toxicity test and number of test organisms that spoils the beverage.

Please address this portion otherwise this article looks like a communication or letters.

Thank you

Author Response

The authors would like to thank reviewer 1 for valuable input and helping to improve the quality of the manuscript. We have included the list of yeasts that were screened against. We have also included some extra figures to show growth inhibition activity at different pH. We were not sure which toxicity tests reviewer 1 was referring to, but if the reviewer can provide more clarity we will gladly try to include more relevant tests if possible.

Reviewer 2 Report

The authors made the recommended corrections.

The quality of the article has improved.

In my opinion, it can be published.

Author Response

Dear Reviewer

Thank you very much for your critical review and for improving the quality of our manuscript. Much appreciated

Regards

BS Chidi

On behalf of authors

Round  3

Reviewer 1 Report

The authors have significantly revised the manuscript and based on the corrections and revisions I feel the manuscript is almost going in forward direction.

Although cytotoxicity studies on the effect of biopreservation compounds have not been reported which can be the study of future scope I would like to know what are the biopreservative compounds produced by the Yeasts. One way could be perform the HPLC and Mass analyses of crude compounds.

Also as i suggested before there is a serious concern on literature reviews. I could not see based on what previous works the research strategy was designed.

Thank you

Author Response

Response to Reviewer Comments

Point 1: The authors have significantly revised the manuscript and based on the corrections and revisions I feel the manuscript is almost going in forward direction.

Response: The authors thank the reviewer for the words of encouragement and willingness to review and improve our manuscript. It is much appreciated

Point 2: Although cytotoxicity studies on the effect of biopreservation compounds have not been reported which can be the study of future scope I would like to know what are the biopreservative compounds produced by the Yeasts. One way could be perform the HPLC and Mass analyses of crude compounds.

Response: As suggested by the reviewer, we listed the potential compounds produced by the biopreservative producing yeasts. The research relating to potential compounds involved in growth inhibition have already been published (see below). The GC-MS method that we used to identify and quantify the compounds are mentioned in that publication.

Mewa-Ngongang, M., du Plessis, H.W., Ntwampe, S.K., Chidi, B.S., Hutchinson, U.F., Mekuto, L. and Jolly, N.P., 2019. Grape Pomace Extracts as Fermentation Medium for the Production of Potential Biopreservation Compounds. Foods, 8(2), p.51.

The focus of the current study was to investigate the activity of the crude biopreservatives and combinations thereof against spoilage yeast consortia. However, there will be a follow up publication where the growth inhibition activity of the selected compounds will be reported.

Point 3: Also as i suggested before there is a serious concern on literature reviews. I could not see based on what previous works the research strategy was designed

Response: Thank you for raising these critical comment. The introduction section has been extensively overhauled to address the reviewer’s concerns